# Selfie-Viewing and Facial Dissatisfaction among Emerging Adults: A Moderated Mediation Model of Appearance Comparisons and Self-Objectification

**DOI:** 10.3390/ijerph17020672

**Published:** 2020-01-20

**Authors:** Jing Yang, Jasmine Fardouly, Yuhui Wang, Wen Shi

**Affiliations:** 1School of Journalism and Communication, Tsinghua University, Beijing 100084, China; yangjing17@mails.tsinghua.edu.cn; 2Centre for Emotional Health, Department of Psychology, Macquarie University, Sydney, NSW 2109, Australia; jasmine.fardouly@mq.edu.au; 3Department of Psychology, Renmin University of China, Beijing 100872, China; 4Department of Earth System Science, Tsinghua University, Beijing 100084, China; shi-w18@mails.tsinghua.edu.cn

**Keywords:** social media, selfies, appearance comparisons, self-objectification, facial dissatisfaction

## Abstract

With the visual turn in online communication, selfies have become common on social media. Although selfies as a way of self-representation provide people with more chances to express themselves, the adverse effects selfies could bring to users’ body image need to be treated seriously. This study tested whether selfie-viewing behaviour on social media was related to facial dissatisfaction and whether appearance comparisons played a mediating role. Moreover, the self-objectification was examined as a moderator between selfie-viewing behaviour and facial dissatisfaction via appearance comparisons. Results showed that more selfie-viewing was associated with higher facial dissatisfaction, and this relationship was mediated by appearance comparisons. The study also found that self-objectification moderated the indirect relation between selfie-viewing and facial dissatisfaction via appearance comparisons. Gender differences were also found to affect the mediation model. Our research provides new insights into the interactions between social media use and perception of body image.

## 1. Introduction

Visual content has become prominent on social media, marking the visual turn in online communication [1,2]. Amongst all forms of visual content circulating online, the selfie has become one of the most ubiquitous practices [3,4], especially among young people [5,6,7]. A selfie is a self-portrait-like photo that is usually taken by oneself, using digital devices, such as smartphones, and then often posted on one’s social media platform [8]. Viewing idealised images in both traditional media (e.g., magazines and billboards) and social media (e.g., Facebook, Instagram, and WeChat) can have a negative influence on individuals’ body image [9,10]. Given that beautification applications installed in smartphones are available to edit and enhance selfies before posting them online, selfies on social media may be considered as idealised images and could thus negatively impact viewers’ body image [11,12]. Therefore, it is necessary to explore the relationship between selfie-viewing on social media (i.e., browsing other people’s selfies and likes/comments attached to them) and body image, as well as the possible mechanisms underlying this relationship.

Notably, focusing on facial appearance is of great significance when conducting research on selfie activities and body image, because selfies primarily contain the user’s face [13]. For example, Hu, Manikonda, and Kambhampati [14] examined the popularity of different image categories on Instagram and found that the most common category was “selfies”, defined as a self-portrait in which only one human face is shown. Furthermore, even though selfie applications are able to edit subjects to have slimmer waists, longer legs, and larger breasts [15], many of the functions selfie applications provide are for facial adjustments, both in terms of the modification of facial features, like larger eyes, as well as comical options, like adding dog ears on a person’s head. Previous research suggests that viewing idealised selfies may especially have the influence on women’s facial concerns, instead of concerns about their physical appearance overall [16]. Therefore, the current study first attempted to investigate the relationship between selfie-viewing and facial dissatisfaction.

### 1.1. Selfie-Viewing and Facial Dissatisfaction

According to the tripartite influence model, media and peers are two of the important and even primary influences on people’s body dissatisfaction and eating disorders, and the other one is family [17]. Selfies posted on social media may combine the effects of media and peers, and thus have a strong influence on users’ body image [10,18]. Like traditional forms of media, selfies may promote an unattainable standard of beauty if they are edited by filters before being uploaded to social media [19]. These editing practices have become normal [20,21], and thus, when viewing other people’s selfies, users are likely to come across a large number of idealised and enhanced images of others. Moreover, unlike traditional media, which only features models and celebrities, social media also features users’ peers or people they know. This is particularly true in a popular Chinese social media WeChat, in which nearly 70% of the contacts are from users’ personal networks, such as family, friends, and co-workers [22,23]. WeChat users could share their updates, including selfies, in *Moment* (the direct translation of the *Moment* in Chinese is “friend circle”), which is similar to Facebook wall [23], and other WeChat contacts could leave comments or just “like” the updates. Thus, based on the tripartite influence model, viewing selfies and the attached comments and likes may have a positive association with people’s facial dissatisfaction.

To our knowledge, only two studies have examined the influence of selfie-viewing behaviour on individuals’ facial dissatisfaction. One research study confirmed that selfie-viewing behaviour contributes to facial dissatisfaction among Chinese adolescents, and internalization of appearance ideals mediated this association [8]. However, this study did not examine the influence of appearance comparisons, which is also suggested as a probable mediating-factor mechanism proposed in the tripartite influence model [17]. The other study, with young women in Australia, used an experimental design and found that viewing other people’s idealised selfies had a negative influence on their facial appearance satisfaction and led them want to change their facial appearance [16]. These studies provide preliminary evidence for how selfie viewing behaviour is associated with facial dissatisfaction. Further research is needed to investigate the relationship between selfie-viewing and facial dissatisfaction, as well as other mechanisms (e.g., appearance comparisons proposed in the tripartite influence model) underlying this association.

### 1.2. Appearance Comparisons as a Mediator

The tripartite influence model also indicates that, between sociocultural sources, namely peers, family, and media, and people’s development of body dissatisfaction, one of the two mechanisms that play the mediating roles is appearance comparisons [17]. Consistent with this theory, studies have supported the mediating role of appearance comparisons in the association between media exposure (including social media use) and body dissatisfaction [24,25,26,27] (Fardouly and Vartanian, 2015; Myers and Crowther, 2007; Slevec and Tiggemann, 2011; Tiggemann and Zaccardo, 2015). Specifically, those who view attractive images of others often judge themselves as being less attractive, which can cause them to feel worse about their appearance [24,28]. Following this rationale, it is possible that appearance comparisons would mediate the relationship between selfie-viewing and facial dissatisfaction.

Firstly, viewing others’ selfies, as well as the likes and comments under the selfies, is likely to trigger appearance comparisons. A study revealed that women regarded the social comparison as a typical driving force that makes them feel bad about themselves after engaging in selfie-related practices [29]. Moreover, Porch [29] figured out that women who reported a higher frequency of viewing selfies often scored higher in social comparisons. In terms of the feedbacks, such as comments and likes, from others regarding selfies posted on social media, this would also intensify the influence that selfies have on making comparisons. To illustrate this point, the number of likes and comments could be seen as a parameter of the attractiveness of the selfies, which strengthens the importance of appearance [8], and people might make comparisons on the likes and comments under different selfies. Second, comparing appearances displayed in selfies may decrease one’s facial satisfaction. As we mentioned above, selfies are usually carefully edited via beatification applications before being posted online. Specifically, in the Chinese context, a report found out that improving selfies by using beautification applications before posting was almost a “must-do” practice among interviewees [30]. Unlike images of celebrities on more traditional forms of media outlets, such as magazines and televisions, selfies on social media may give viewers an illusion that the glamorised faces are real and attainable, neglecting the fact that these selfies can be enhanced by the filters [29]. Consequently, when viewing selfies, social media users may judge themselves to be less attractive than others (i.e., make upward comparisons, Festinger, 1954), which may result in facial dissatisfaction [28,31,32]. Thus, appearance comparisons would mediate the positive relationship between the frequency of selfie-viewing behaviour and the extent of facial dissatisfaction.

### 1.3. Self-Objectification as a Moderator

The extent to which appearance comparisons influence the link between selfie-viewing behaviour and facial dissatisfaction may depend on a person’s level of self-objectification [33,34]. According to objectification theory, self-objectification can be seen as both a stable personal trait and a context-dependent state [35,36], and in the current study, we conceptualise self-objectification as a stable trait. When we take a specific look at the trait self-objectification; we could argue that when being exposed to sexually objectifying images (e.g., thin ideals and other images that focus overtly on appearance), people may come to take an observers’ perspective of themselves and treat themselves as an “object” to be stared at [35,37]. Therefore, people who are higher on self-objectification pay more attention to their physical appearance than other aspects of themselves. Numerous studies find that self-objectification can intensify the impact of sexually objectifying experiences on body image. [38]. Similarly, self-objectification may moderate the indirect link between selfie-viewing behaviour and satisfaction of appearance via appearance comparisons.

First, self-objectification might moderate the path between selfie-viewing behaviour and appearance comparisons. Lindner, Tantleff-Dunn, and Jentsch [39] suggest that self-objectified individuals are more likely to make appearance comparisons with others. The more one puts value on physical appearance, the more likely they are to make appearance comparisons with others. The positive relation between self-objectification and appearance comparisons has also been supported by some empirical research [39,40]. Additionally, a recent study found that self-objectification moderated the association between social media usage and appearance comparisons, such that compared with people with low levels of self-objectification, those with high self-objectification engaged in more appearance comparisons when using social media [41]. Following this rationale, self-objectification might moderate the link between selfie-viewing behaviour and appearance comparisons.

In addition, the path from appearance comparisons to facial dissatisfaction may also be moderated by self-objectification. Previous studies have shown that self-objectification is associated with body dissatisfaction [25,42,43,44]. Moreover, as Quinn, Chaudoir, and Kallen [45] stated, social comparison serves as a way for self-objectified individuals to monitor their appearance. In other words, though appearance comparisons might cause people’s facial dissatisfaction, the degree of impact may differ depending on their levels of self-objectification. Through the process of comparison, people become aware of the disparity between themselves and the comparison targets. Subsequently, self-objectified individuals who are more concerned about physical appearance may feel more dissatisfied if they judge themselves to be worse off than the target.

### 1.4. Considerations about Gender

The majority of research on media exposure, especially social media use, and body image has focused on women, but scientists are achieving more consensus that these factors are also important for men [46]. Previous research found that there was no gender difference in the experience of body dissatisfaction after being exposed to idealised images on social media [47], and in the relationship between social media use and body dissatisfaction [48].

Of particular relevance to the present study, existent studies show that women tend to take more selfies than men [3,49,50,51] and feel worse after engaging in appearance-related comparisons [50]. However, according to Liziliziduanxin [52], there is a growing number of males using photo-beautification apps, which may be representative of males’ facial concerns. A recent study also showed that no gender difference was found in the indirect relationship between selfie-viewing and facial dissatisfaction via general attractiveness internalisation among adolescents [8]. Thus, the present study aimed to investigate the moderating role of gender in the mediation model, in which selfie-viewing has an indirect effect on facial dissatisfaction through appearance comparisons.

### 1.5. The Current Study

To sum up, this study aimed to investigate (1) whether selfie-viewing behaviour would be positively correlated to the extent of individual’s facial dissatisfaction, (2) whether appearance comparisons would mediate the correlation between selfie-viewing behaviour and facial dissatisfaction, (3) whether the indirect link between selfie-viewing behaviour and individuals’ facial dissatisfaction through appearance comparisons would be moderated by self-objectification (moderated mediation model, see Figure 1), and (4) the role of gender in the mediation model. Based on the tripartite influence model and literature reviews mentioned above, we put forward the following hypotheses. Selfie-viewing behaviour would be positively correlated to individuals’ facial dissatisfaction (Hypothesis 1). Appearance comparisons would mediate the relationship between selfie-viewing behaviour and individuals’ facial dissatisfaction (Hypothesis 2). Self-objectification would moderate the indirect links between selfie-viewing behaviour and individuals’ facial dissatisfaction via appearance comparisons (Hypothesis 3). Considering the inconsistency of previous findings and the lack of studies focusing on men, gender was investigated as an explorative variable in the present study.

## 2. Methods

### 2.1. Participants

Participants consisted of 481 college students from in China, of which 281 (58%) were females and 200 (42%) were males. The age of participants was between 17 to 22, where the average was 19.44 years (the SD was 1.18), and the mean body mass index (by kg/m^2^) was 20.68 (the SD was 2.76).

### 2.2. Measures

Selfie-Viewing

Following previous research (Y. Wang et al., 2019), selfie-viewing was measured by three items. One item was used to assess participants’ frequencies of viewing selfies on social media [53]. Participants could choose their options from 1 (viewing selfies *very infrequently)* to 6 (viewing selfies *several times a day)*. The second and third items assessed the extent to which individuals look selfies of other people on social media [54,55]. Two 7-point scale (1= *not at all*, 7= *very much*) were adopted when measuring these times. As different response ranges were adopted when measuring the three items, we standardised all the items with *z*-scores. We chose mean scores as the measurement, where the higher scores referred to greater frequency of selfie-viewing on social media. Cronbach’s alpha was 0.75 in this study.

### 2.3. Appearance Comparisons

An updated version of the Physical Appearance Comparisons Scale (PACS) [17], developed by Fardouly and Vartanian [24], was used to assess participants’ tendency to compare their appearance with the appearance of others on social media. Participants were required to report to what extent they agree with three items (e.g., “When using social media, I make comparisons between my physical appearance and other individual’s physical appearance”) on a 5-point scale (1 = *definitely disagree*; 5 = *definitely agree*). Items were averaged to form a total score, with a higher score indicating a higher tendency to make comparison between one’s appearance and other individuals’ appearance on social media. Cronbach’s α was 0.84 in this study.

### 2.4. Self-Objectification

A Self-Objectification Questionnaire (SOQ) [56] was introduced to examine the extent of self-objectification. Participants were instructed to give a rank order of to what extent ten attributes are to their physical self-concept, from 0 (*least important*) to 9 (*most important*). These ten attributes consisted of not only five competence-related attributes (i.e., health, physical condition, physical fitness level, strength, and energy level), but also five attributes concerning appearance (i.e., sex appeal, weight, physical attractiveness, body measurements, and firm or sculpted muscles). To calculate the final score on the questionnaire, the sum of the competence items was subtracted from the sum of the appearance items. Final scores could range between −25 and 25; when scores were higher, self-objectification was greater.

### 2.5. Facial Dissatisfaction

As for the assessment of facial dissatisfaction, we used the Facial Appearance Concern (FAC) subscale of the Negative Physical Self Scale (NPSS) [57]. The FAC subscale has 11 items. Participants responded to each item on a 5-point scale, ranging from 0 = *never* to 4 = *always*. A representative item was “I am depressed about how my face looks”. Items were first averaged and then summed to get a total score, where a higher score referred to greater extent of facial dissatisfaction. The FAC subscale has both a stable factor structure and good reliability and validity [57]. As for the present study, the measure demonstrated good internal reliability (α = 0.92).

### 2.6. Procedure

Ethical approval was gained from the Ethics Committee of the corresponding author’s University. The participants were recruited by an advertisement about mental health and Internet use. Before the survey was carried out, privacy concerns, freedom of withdraw, and anonymity of our study were discussed with the participants, and we were sure that all participants showed their agreement. Well-trained assistants were invited to assist in the research. Participants were required to fill out a list of questions concerning demographic information, selfie-viewing, appearance comparisons, facial dissatisfaction, and self-objectification.

## 3. Results

### Basic Analyses

The proportion of missing data was less than 2%, and the missing part was tackled via pairwise deletion. Statistics for all variables are presented in Table 1. We can tell that selfie-viewing behaviour was positively correlated to appearance comparisons behaviour and the extent that individuals are unsatisfied with their appearance. Appearance-comparisons behaviour was positively correlated to the self-objectification degree and facial-dissatisfaction degree. Self-objectification was positively associated with the extent that individuals are unsatisfied with their appearance. The association between selfie-viewing and self-objectification was nonsignificant. Since selfie-viewing was positively related to the extent that individuals are unsatisfied with their appearance, our Hypothesis 1 was valid.

## 4. Testing for Mediation Effect

In Hypothesis 2, this study proposed that appearance comparisons would mediate the link between selfie-viewing and facial dissatisfaction. To test this hypothesis, PROCESS macro (Model 4) in SPSS [58] was conducted. Age and BMI were entered as covariates in the analysis. As Table 2 illustrates, selfie-viewing positively predicted facial dissatisfaction, *b* = 0.18, *p* < 0.001 (Model 1). Selfie-viewing behaviour was positively linked with appearance comparisons, *b* = 0.34, *p* < 0.001 (Model 2), and appearance comparisons behaviour was positively related to the extent of facial dissatisfaction, *b* = 0.42, *p* < 0.001 (Model 3).

The indirect effect of selfie-viewing frequency on the extent of facial dissatisfaction via appearance comparisons was 0.14 (*SE* = 0.02, 95% CI = [0.10, 0.19]). Empirical 95% CI did not include zero, indicating that selfie-viewing exerted a significant indirect effect on facial dissatisfaction through appearance comparisons. This indicated appearance comparisons mediated the links between selfie-viewing behaviour and the extent of facial dissatisfaction. Therefore, Hypothesis 2 was regarded as valid.

### 4.1. Testing for Moderated Mediation

In Hypothesis 3, we expected that self-objectification would moderate the indirect relations between selfie-viewing and facial dissatisfaction via appearance comparisons. To examine the moderated mediation hypothesis, we estimated parameters with the PROCESS macro (Model 58) by Hayes [58]. Specifically, we investigated the moderating effect of self-objectification on the relationships between: (1) selfie-viewing behaviour and appearance-comparisons behaviour; and (2) appearance comparisons and the extent of facial dissatisfaction. Age and BMI were entered as covariates in the analysis.

As Table 3 illustrates, selfie-viewing was positively associated with appearance comparisons (*b* = 0.37, *p* < 0.001). However, the interplay between selfie-viewing and self-objectification on appearance comparisons was nonsignificant, *b* = 0.03, *p* = 0.50 (Model 1). This indicates that the relationship between selfie-viewing and appearance comparisons was not moderated by self-objectification. In addition, appearance comparisons had a positive association with facial dissatisfaction, *b* = 0.39, *p* < 0.001 (Model 2), and this association was moderated by self-objectification, *b* = 0.12, *p* < 0.01. To facilitate the interpretation of this interaction effect, we plotted predicted facial dissatisfaction by appearance comparisons, separately for low and high self-objectification (1 SD below the mean and 1 SD above the mean, respectively) (Figure 2). Simple slope tests indicated that, for individuals who had higher levels of self-objectification, appearance comparisons were positively and significantly associated with facial dissatisfaction, *b*
_simple_ = 0.49, *p* < 0.001. For those with lower levels of self-objectification, the association between appearance comparisons and facial dissatisfaction was still significant but much weaker, *b*
_simple_ = 0.25, *p* < 0.001.

The bias-corrected percentile bootstrap results further demonstrated that the indirect effect of selfie-viewing on the extent of facial dissatisfaction via appearance comparisons was moderated by self-objectification. For individuals with high self-objectification in particular, the indirect effect of selfie-viewing behaviour on the extent of facial dissatisfaction was significant, *b* = 0.21, *SE* = 0.05, 95% CI = [0.12, 0.33]. This indirect effect was still significant but much weaker for those low in self-objectification, *b* = 0.09, *SE* = 0.03, 95% CI = [0.04, 0.17]. Given that self-objectification only moderated the second stage of the mediation process, namely the path from appearance-comparisons behaviour to the extent facial dissatisfaction, Hypothesis 3 was partially valid.

### 4.2. Examining Gender Difference

Table 4 displays the means and standard deviations variables for men and women separately. Independent samples *t*-tests revealed that women reported higher selfie-viewing frequency than men. There was no significant difference in appearance comparisons, self-objectification, and facial dissatisfaction between genders.

As for the moderating role that gender might have in the mediation model, we analysed it by using PROCESS macro (Model 59) by Hayes [58]. According to Table 5, the interplay between selfie-viewing behaviour and gender on appearance-comparisons behaviour was not significant. Furthermore, the association between selfie-viewing and gender on facial dissatisfaction was nonsignificant, while the interaction between appearance comparisons and gender on facial dissatisfaction was significant. That is, gender moderated the relationship between appearance comparisons and facial dissatisfaction. In considering of the descriptive purpose, we plotted the predicted facial dissatisfaction by appearance comparisons separately for males and females (Figure 3). Simple slope tests pointed out that, for males, higher levels of appearance comparisons were related to higher levels of dissatisfaction toward appearance, *b*
_simple_ = 0.52, *p* < 0.001. However, as for females, the association between appearance comparisons and facial dissatisfaction was still significant, but much weaker, *b*
_simple_ = 0.32, *p* < 0.001.

## 5. Discussion

Based on the tripartite influence model [17], this study inspected the links between selfie-viewing behaviour on social media and dissatisfaction toward appearance in young adults. We also evaluated the mediating role of appearance comparisons in this relationship. The findings of this study showed that selfie-viewing behaviour was positively associated with the extent of facial dissatisfaction and that this link was mediated by appearance comparisons. We also inquired into the moderating role that self-objectification may have in the mediation model, and we found that self-objectification moderated the link from appearance comparisons to facial dissatisfaction. Last but not the least, we tested the potential moderating role of gender in the mediation model and found that gender moderated the path from appearance comparisons to facial dissatisfaction.

### 5.1. Selfie-Viewing and Facial Dissatisfaction

Consistent with our hypothesis, the present study discovered that selfie-viewing frequency positively correlated with the extent of facial dissatisfaction among young adults. This finding echoes previous studies, suggesting that beauty ideals presented on both traditional media and social media may affect body image [10,34,59] as well as the recent research on the association between selfie-viewing behaviour and the extent of facial dissatisfaction conducted in adolescents [8]. An important explanation for this result is that selfies on social media may promote an unattainable standard of beauty because almost all of these selfies are enhanced before being posted on social media. Taking China for example, almost everyone would spend much time, about 40 min per face, according to a report [30], to edit their selfies before posting them online. As a result, viewing these selfies would increase the viewers’ facial dissatisfaction. Furthermore, findings of this study provided a complement for previous literature, suggesting that social media use, and selfies in particular, is related to facial appearance concerns [8,16,37]. This suggests that future research should examine users’ facial satisfaction in the context of social media use.

### 5.2. Appearance Comparisons’ Mediation Role

Our findings also showed that appearance comparisons mediated the relationship between selfie-viewing and facial dissatisfaction, which is consistent with the findings of previous research on the relationship between social media use and body image concerns [7,27]. As the tripartite influence model depicts, appearance comparisons is a critical mediating factor behind the link from sociocultural influence and body dissatisfaction [17]. Extending this model, our study first examined the mediating role of appearance comparisons in the relationship between selfie-viewing frequency and the extent of facial dissatisfaction, which adds further empirical support for aspects of the tripartite influence model in the social media era and extends the model by focusing on dissatisfaction with facial appearance rather than the whole body.

Additionally, the two stages in the mediation model require of our attention. First, selfie-viewing was positively correlated to appearance comparisons, which is in accordance with previous findings that individuals seeing selfies more frequently showed higher levels of appearance comparisons than those seeing less frequently [29]. This finding can be explained by the tripartite influence model [60]. According to this model, peers and media are important sources for appearance comparisons, and selfies on social media could be seen as a combination of media and peer pressures, for they are a type of media that can be generated and posted by peers. Furthermore, the comments and likes received by the selfies may magnify the impact of selfie-viewing on appearance comparisons. Specifically, if you notice that other people endorse the attractiveness of the person in the selfie by giving positive comments or likes, it may increase the salience of the person’s appearance and lead to more frequent appearance comparisons. Second, the path from appearance comparisons to facial dissatisfaction was also significant. An important interpretation for this result is the fact that selfies on social media are usually heavily improved. Given the edited nature of many selfies on social media, users tended to devalue themselves as less attractive compared with the person in the image (i.e., make upward comparisons), which may lead to dissatisfaction toward appearance. Furthermore, because people on social media are likely to be perceived as peers, viewers may not be as aware of the unrealistic and unattainable nature of the images as they may be of images in magazines, and this may enhance the impact of comparisons. [61]. Notably, the present study did not specifically focus on peers’ selfies and comparisons with peers. Future research investigating selfie-viewing and facial dissatisfaction should pay more attention to the influence of peers on social media.

### 5.3. The Moderating Role of Self-Objectification

We also investigated the moderating role of self-objectification in the indirect relationship between selfie-viewing and facial dissatisfaction via appearance comparisons. Consistent with our hypothesis, self-objectification moderated the path from appearance comparisons to facial dissatisfaction. Specifically, the relationship between appearance-comparisons behaviour and facial dissatisfaction was stronger for those with higher value of self-objectification than those with lower value of self-objectification. This finding suggests that people may vary in the extent to which they are influenced by appearance comparisons, which is in accordance with the previous literature on exposure to idealised images [62]. According to objectification theory, self-objectified individuals regard their appearance-based attributes as more important than their functional attributes [35]. If people high in self-objectification place more value on their physical appearance, they may feel worse when they perceive another person to be more physically attractive than them.

Opposed to our hypothesis, the moderating role of self-objectification was not discovered in the path from selfie-viewing to appearance comparisons, which conflicts with previous finding that self-objectification moderated the link from social media use and appearance comparisons [41]. One possible explanation for this null result might be that selfies mainly focus on facial features, and thus the comparisons after viewing selfies are also facial comparisons. However, in terms of the measure of self-objectification in the present study, the appearance-related attributes in the scale are almost about body shape or size. Therefore, no moderating of self-objectification was observed in the relationship between selfie-viewing and appearance comparisons. Another explanation for this finding might be that appearance comparisons may be automatic and thus individuals are hardly able to be aware of this process [63]. Of course, considering that this is an unprecedented study, examining the moderating role of self-objectification in the link from selfie-viewing to appearance comparisons, more research in the future is needed to better understand the mechanisms.

### 5.4. The Role of Gender

Our results indicated that there was no significant gender difference in appearance comparisons, self-objectification, and facial dissatisfaction. This finding may partly be due to males becoming more concerned about their physical appearance, as a growing number of males in China are purchasing beauty products and wearing makeup [64]. Moreover, previous empirical research supports the lack of gender difference in facial appearance concerns [8,65]. Our results showed that gender was not a moderate factor in terms of the relationships between selfie-viewing behaviour and appearance-comparisons behaviour, or between selfie-viewing behaviour and the extent of facial dissatisfaction. These findings indicated that, like females, males may engage in appearance comparisons and feel unconfident about their facial appearance after viewing others’ selfies. Therefore, appearance comparisons may be automatic for both women and men when they view selfies, which may lead them to feel dissatisfied with their facial appearance.

Gender did, however, have a moderating function in the relation between appearance comparisons and dissatisfaction toward the face, such that the relationship was stronger for men than for women. A possible explanation for this finding is the “drench” effect proposed by Greenberg (1988). As opposed to cultivation theory, which emphasises the media’s day-by-day subtle influence on individuals’ perceptions and behaviours, Greenberg [66] argues that an extraordinary, uncharacteristic, and sudden stimulus would have a heavier effect on individuals’ beliefs. In terms of the current study, facial concerns have recently been emphasised in China as important for men’s attractiveness, which has previously primarily been portrayed as being important for women. Hence, facial comparisons may be more novel for men, and thus the same levels of comparisons may have a stronger effect on them. However, this is only one speculation we offer. More follow-up research is needed to figure out the possible gender difference in the relationship between appearance comparisons and facial dissatisfaction.

### 5.5. Limitations and Further Research

The limitations of the current study should be acknowledged. First, the current study only focused on college students. Further studies could pay attention to different age groups. For example, pre-adolescents, who are at the start of their primary schooling, are at a critical stage of experiencing body dissatisfaction [67], and the age that begins to use smartphones is decreasing [68,69]; thus, selfies that users of this young age group are exposed to may insert significant influence on their development of body image [70]. Second, the present study was cross-sectional; thus, the direction of the relationships cannot be determined. For example, it is also possible that those high in facial dissatisfaction and self-objectification make more comparisons on social media or engage with more selfies than others. Further temporal dimension is necessary for testing the directions of these relationships. The third limitation concerns the measure in this study. We did not ask specifically whether the selfies being viewed were of the users’ friends, family members, or celebrities, which needs to be addressed further, because comparisons to known peers may differ from those to celebrities [24]. In addition, the present study used a measure of appearance comparisons in general rather than facial appearance comparisons. Future research should focus on facial appearance comparisons when conducting research on selfie activities and facial dissatisfaction. Fourth, the present study focused on social media generally and did not distinguish between different platforms. Different platforms have different affordances that may shape users’ behaviours accordingly [5,71]. For instance, as mentioned above, WeChat is a social media platform mainly for Chinese people to connect with contacts in real life, in which most of them might be families, friends, and colleagues [22,23], whereas Sina Weibo is a social media platform containing users from various social classes and occupations, including both the public and celebrities, in China. Thus, the selfies uploaded to these two platforms are likely to be different, which may in turn affect individuals’ facial satisfaction in different ways. Besides, short-video platforms, such as *Douyin* (its overseas version is *Tik Tok*), where users conduct webcasts and upload short videos rather than still images, are able to detect and tailor the performers’ bodies automatically, for example, making legs longer and eyes bigger in sync with the webcast [72]. Compared with images, videos provide users with a stronger sense of presence [73], creating an atmosphere that audiences are together with the video blogger in the same space, without any mediation [74], which may have a stronger influence on audiences, including on their body image.

### 5.6. Conclusions

Based on the tripartite influence model, this study examined the links between selfie-viewing behaviour and the extent of facial dissatisfaction, as well as the potential mechanisms underlying this relationship. Our results showed that selfie-viewing behaviour was associated with facial dissatisfaction, and this relationship was mediated by appearance comparisons. These findings extend the extant literature and the tripartite influence model and contribute to a better understanding of the influence of social media use on body image. Furthermore, for those high in self-objectification, the indirect correlation from selfie-viewing behaviour to the extent of facial dissatisfaction via appearance comparisons was stronger than that of those with low self-objectification. Our results provide theoretical and practical implications for researchers, that reducing individuals’ self-objectification by instructing people to focus more attention on function or competence-related attributes of body rather than physical-appearance-related ones would be helpful in buffering the negative effects of selfie-viewing on body image. In addition, our results showed that gender moderated the mediation model, suggesting that the link between comparisons and facial dissatisfaction may be stronger for male than females. More research is needed, especially using longitudinal designs, to further explore these relationships and any potential reasons for gender differences in the relationship between selfie activities and appearance concerns.

## Figures and Tables

**Figure 1 ijerph-17-00672-f001:**
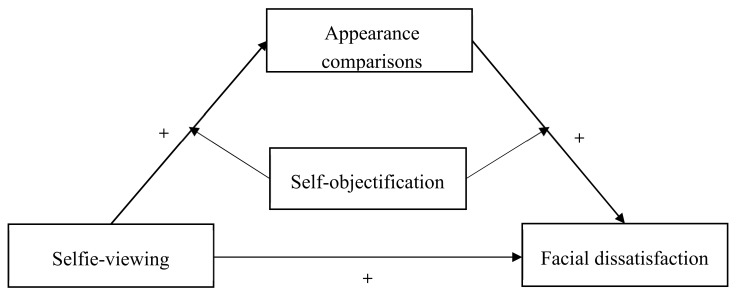
The proposed moderated mediation model.

**Figure 2 ijerph-17-00672-f002:**
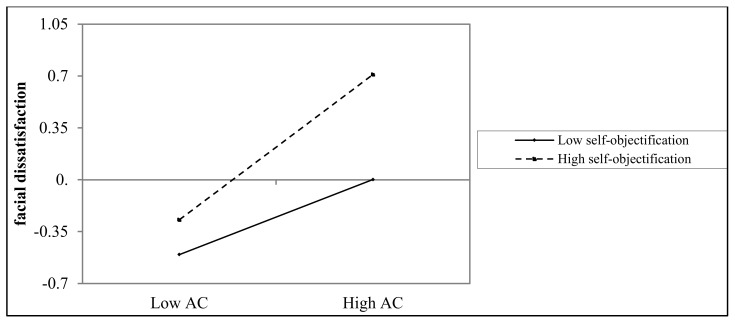
The interaction between appearance comparisons and self-objectification on facial dissatisfaction.

**Figure 3 ijerph-17-00672-f003:**
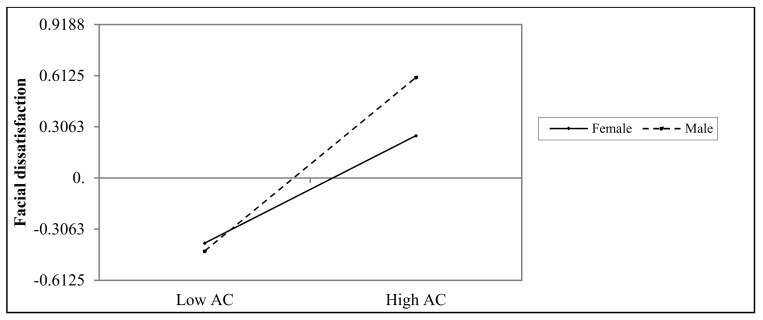
The interaction between appearance comparisons and gender on facial dissatisfaction. AC = appearance comparisons.

**Table 1 ijerph-17-00672-t001:** Means, standard deviations, and zero-order correlations for main study variables.

Variables	M	SD	1	2	3	4
1. Selfie-viewing	0	0.76	1			
2. Appearance comparisons	2.14	0.84	0.34 **	1		
3. Self-objectification	−9.64	10.05	0.04	0.20 **	1	
4. Facial dissatisfaction	1.14	0.72	0.17 **	0.43 **	0.33 **	1

Note: Mean scores for selfie-viewing were calculated by averaging *z*-scores of all items. ** *p* < 0.01.

**Table 2 ijerph-17-00672-t002:** Testing the mediation effect of selfie-viewing on facial dissatisfaction.

Predictors	Model 1 (Facial Dissatisfaction)		Model 2 (Appearance Comparisons)		Model 3 (Facial Dissatisfaction)	
	*b*	*t*	*b*	*t*	*b*	*t*
BMI	0.1	20.15 *	0.05	10.19	0.08	10.82
Age	0.01	0.35	−0.02	−00.52	0.02	0.61
Selfie-viewing	0.18	30.88 ***	0.34	70.72 ***	0.03	0.75
Appearance comparisons					0.42	90.63 ***
*R* ^2^	0.04		0.12		0.2	
*F*	60.52 ***		200.32 ***		290.04 ***	

Note: * *p* < 0.05, *** *p* < 0.001.

**Table 3 ijerph-17-00672-t003:** Testing the moderated mediation effect of selfie-viewing on facial dissatisfaction.

Predictors	Model 1 (Appearance Comparisons)		Model 2 (Facial Dissatisfaction)	
	*b*	*t*	*b*	*t*
BMI	0.07	10.42	0.08	10.6
Age	−0.01	−0.21	0.03	0.59
Selfie-viewing (SV)	0.37	70.43 ***	0.02	0.47
Self-objectification (SO)	0.19	30.78 ***	0.24	40.79 ***
SV × SO	0.03	0.68		
Appearance comparisons (AC)			0.39	70.39 ***
AC × SO			0.12	20.80 **
*F*	140.85 ***		220.74 ***	
*R*²	0.19		0.31	

Note. ** *p* < 0.01. *** *p* < 0.001.

**Table 4 ijerph-17-00672-t004:** Means (SD), and independent samples *t*-tests comparisons between males and females.

Variables	M (SD)		*t*	*d*
	Male	Female		
Selfie-viewing	−0.12 (0.83)	0.09 (0.69)	3.06 **	0.29
Appearance comparisons	2.14 (0.88)	2.13 (0.81)	0.16	0.01
Self-objectification	−10.03 (10.36)	−9.38 (9.86)	0.57	0.06
Facial dissatisfaction	1.20 (0.75)	1.10 (0.70)	1.6	0.13

Note: Mean for selfie-viewing was calculated by averaging *z*-scores of all items. ** *p* < 0.01.

**Table 5 ijerph-17-00672-t005:** Testing the moderating role of gender in the mediation model.

Predictors	Model 1 (Appearance Comparisons)		Model 2 (Facial Dissatisfaction)	
	*b*	*t*	*b*	*t*
Selfie-viewing (SV)	0.39	60.23 ***	0.08	10.24
Gender	0.11	10.25	0.15	10.81
SV × gender	−0.08	−0.93	−0.07	−0.78
Appearance comparisons (AC)			0.32	50.42 ***
AC × gender			0.2	20.27 *
*F*	210.44 ***		230.73 ***	
*R*²	0.12		0.2	

Note: Gender was dummy coded such that male = 1 and female = 0. * *p* < 0.05. *** *p* < 0.001.

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
