# Peer review of "Selfie-Viewing and Facial Dissatisfaction among Emerging Adults: A Moderated Mediation Model of Appearance Comparisons and Self-Objectification"

_ijerph, 2020, doi:10.3390/ijerph17020672_

Round 1

Reviewer 1 Report

The authors propose the questionnaire to support their hypothesis. For questionnaire, it consists of 281 girls and 200 boys. What happens if we change the proportion (i.e., 400 girls and 100 boys or vice verse). This is because the girls prefer to selfie. Since the authors focus on social medial, why do not use the computer to collect the information of selfie and do prediction. For example:"Virtual makeover: Selfie-taking and social media use increase selfie-editing frequency through social comparison" and "Prediction of collective action using deep neural network and species competition model on social media" and "THE RELATIONSHIP BETWEEN APPEARANCE CONCERNS AND SELFIE SHARING ON SOCIAL MEDIA". The format needs to be improved. For example, pp.5, 8, 10. A page just holds a figure.

Author Response

Reviewer 1:

Comments to the Author:

The authors propose the questionnaire to support their hypothesis. For questionnaire, it consists of 281 girls and 200 boys. What happens if we change the proportion (i.e., 400 girls and 100 boys or vice versa). This is because the girls prefer to selfie. Since the authors focus on social media, why do not use the computer to collect the information of selfie and do prediction. For example: “Virtual makeover: Selfie-taking and social media use increase selfie-editing frequency through social comparison" and "Prediction of collective action using deep neural network and species competition model on social media" and "THE RELATIONSHIP BETWEEN APPEARANCE CONCERNS AND SELFIE SHARING ON SOCIAL MEDIA". The format needs to be improved. For example, pp.5, 8, 10. A page just holds a figure.

RESPONSE: Thank you very much for your feedback. In the following section, you will find our responses to each of your comments.

The authors propose the questionnaire to support their hypothesis. For questionnaire, it consists of 281 girls and 200 boys. What happens if we change the proportion (i.e., 400 girls and 100 boys or vice versa). This is because the girls prefer to selfie.

RESPONSE: You have mentioned a very important question regarding the gender difference in selfie-related behaviors and the following consequences. There is an increasing awareness of the phenomenon that there is no gender difference in terms of the development of gender body dissatisfaction and after being exposed to idealised images on social media (Haferkamp & Krämer, 2011). Besides, in the Chinese context, there is a growing number of males using photo-beautify apps. Out of the particular concern on gender difference, our study has taken gender difference into consideration, in which we paid a specific look at the moderating role of gender in the mediation model (Page 4). Thus, we think that in our study, striving to make a balance between the number of male and female participants is necessary. As for the advice you raised in the comments about changing the proportion of the participants number of different genders, we think it a valuable starting point for future study. Thank you very much.

Since the authors focus on social media, why do not use the computer to collect the information of selfie and do prediction. For example: “Virtual makeover: Selfie-taking and social media use increase selfie-editing frequency through social comparison" and "Prediction of collective action using deep neural network and species competition model on social media" and "THE RELATIONSHIP BETWEEN APPEARANCE CONCERNS AND SELFIE SHARING ON SOCIAL MEDIA".

RESPONSE: Thank you for your comment. We have carefully read this part of the comments and discussed deeply about the suggestions you raised in this comment. Also, three articles you listed were well read after we receiving the feedback. And finally, we came up with the following responses. First of all, in terms of choosing offline questionnaire to collect data rather than online survey, we think that though online survey provides us with more convenient way in data collection like one of the studies you mentioned (Chae, 2017), offline survey has some advantages in making sure the quality of the answers (Cohen et al. 2018; Lonergan et al., 2019). Also, by conducting offline survey, the demographic requirements of the participants the current study is concerned can be controlled in some extent. Regardless of financial constraints or other limitations, doing offline survey is still a conventional and decent approach to collect data, especially for those involves individual-level information and personality tests. The other study you mentioned (Seyfi & Arpaci, 2016) also conducted offline survey.

Our study aims at figuring out one of the possible underlying mechanisms between selfie-viewing and facial dissatisfaction. However, your advice in using deep neutral network and other computational methods to make predictions on behaviours is insightful and practical (Yang et al., 2019). We think our study could be one of the cornerstones in assisting future studies relating to behaviour prediction.

The format needs to be improved. For example, pp.5, 8, 10. A page just holds a figure.

RESPONSE: Thank you so much for pointing out the format issues. We have modified the format and made sure that only one figure or table is contained in one page (Page 5, 7, 8, 9, 10, 11, 12, 13).

Reviewer 2 Report

This study investigated the effects of selfie-viewing on social media on facial dissatisfaction through appearance comparison. The moderation effect of self-objectification was also examined. I found some issues that the authors need to address in their manuscript.

First and the most important, I am wondering why self-objectification is not a mediator, but a moderator, of the association between selfie-viewing and facial dissatisfaction. I would expect that selfie-viewing should be associated with self-objectification, as indicated by the objectification theory. Though the results showed that selfie-viewing was not significantly associated with self-objectification, it might be that the objectification measure was more body-related rather than facial-related as what the authors mentioned.

Generally, self-objectification should be conceptualized as a stable individual factor to be a moderator. However, it seems to me that the current conceptualization was more contextual based, aligning with the objectification theory. Therefore, the author should clearly justify why self-objectification was conceptualized as a moderator of the association between selfie-viewing and facial dissatisfaction.

In addition, the authors should provide more details on the tripartite influence model.

Second, the author did not make clear whether their measurement of selfie-viewing was related to the selfie-viewing on social media. The author should clearly indicate this information when describing the scale. If not, then the contextual base of this paper might need to be changed.

Third, regarding the analyses, were demographic variables and BMI controlled in the regression analyses? If not, the authors should consider those variables as covariates in the regression models. Also, in table 3, the value of “SV´SO” was missing.

Final, there are a number of wording mistakes that need to be corrected, especially in the method and result section. The authors should carefully proofread the paper. Following are some examples:

P6, selfie-viewing:

“On item” --> one item;

“measuring these times” --> measuring these items;

“different response ranges are adopted” --> different response ranges were adopted

Author Response

Reviewer 2:

Comments to the Author:

This study investigated the effects of selfie-viewing on social media on facial dissatisfaction through appearance comparison. The moderation effect of self-objectification was also examined. I found some issues that the authors need to address in their manuscript.

RESPONSE: Thank you very much for your feedback. In the following section, you will find our responses to each of your comments.

First and the most important, I am wondering why self-objectification is not a mediator, but a moderator, of the association between selfie-viewing and facial dissatisfaction. I would expect that selfie-viewing should be associated with self-objectification, as indicated by the objectification theory. Though the results showed that selfie-viewing was not significantly associated with self-objectification, it might be that the objectification measure was more body-related rather than facial-related as what the authors mentioned.

Generally, self-objectification should be conceptualized as a stable individual factor to be a moderator. However, it seems to me that the current conceptualization was more contextual based, aligning with the objectification theory. Therefore, the author should clearly justify why self-objectification was conceptualized as a moderator of the association between selfie-viewing and facial dissatisfaction.

RESPONSE: Thank you so much for commenting on one of the key variables of our study. This gave us a chance to reflect on the very basic of the idea of self-objectification. We agree with your opinion that self-objectification might function as a mediator in the relationship between selfie-viewing and facial dissatisfaction. We treated self-objectification as a moderator in our study for the following reasons: First, we built our mediating model based on the tripartite influence model, which clearly states the mediating role appearance comparisons plays in the association between sociocultural influence (e.g., media and peers) and body image. Second, self-objectification was treated as moderator in many previous studies in the field of body image and eating disorders. For example, self-objectification moderated the relationship between sexual objectification experiences and eating disorders (Moradi & Huang, 2008), as well as the relationship between selfie-related behaviours and eating disorders (Cohen et al., 2018). Third, like you mentioned in the comment, self-objectification concerned more about the whole body instead of solely the facial appearance, it might fail to mediate the relationship between selfie-viewing and facial dissatisfaction. However, for individuals with high self-objectification, physical appearance (including facial appearance) is very important and thus the indirect relationship between selfie-viewing and facial dissatisfaction through appearance comparisons might be stronger for them. In other words, self-objectification would moderate this indirect relationship.

In addition, as per your suggestion, we have conceptualized self-objectification as a stable individual factor to be a moderator in the indirect relationship between self-viewing and facial dissatisfaction in the revised manuscript (Page 3).

We searched and read more research articles and found out that self-objectification can both be used as a mediator and a moderator. Several studies have found the moderating role that self-objectification played in the relationship between sexual objectification experiences and eating disorders (Moradi & Huang, 2008), as well as between selfie-related behaviours and eating disorders (Cohen et al., 2018). Besides, we built our mediating model based on the tripartite influence model, which clearly states the mediating role appearance comparisons plays between media and peers, and body image. Next, we tried to discuss the possible moderating role that self-objectification might play. In the meanwhile, like you mentioned in the comment, self-objectification concerned more about the whole body instead of solely the facial appearance, it might fail to mediate the relationship between selfie-viewing and facial dissatisfaction. Thus, in our study, we treat self-objectification as a moderator. And we justified the role self-objectification may play in two separate paths, the first one is that self-objectification might moderate the path between selfie-viewing behaviour and appearance comparisons. And the second is that the path from appearance comparisons to facial dissatisfaction may also be moderated by self-objectification.

In addition, in terms of the feature of self-objectification, we found that it is conceptualized as both a trait and a context-dependent state. And we have added explanation to this in the revised manuscript which is highlighted in yellow color (Page 3). Thank you very much for bringing up this issue.

In addition, the authors should provide more details on the tripartite influence model.

RESPONSE: Thank you for this valuable comment and your time in carefully reading our manuscript. We have added more information about tripartite influence model in literature review section (Page 2) and highlighted them using yellow color.

Second, the author did not make clear whether their measurement of selfie-viewing was related to the selfie-viewing on social media. The author should clearly indicate this information when describing the scale. If not, then the contextual base of this paper might need to be changed.

RESPONSE: Thank you for this valuable comment. As per your suggestion, we updated the description of the measurement of selfie-viewing in the revised manuscript to indicate that selfie-viewing is related to the selfie-viewing on social media. Please refer to Page 6.

Third, regarding the analyses, were demographic variables and BMI controlled in the regression analyses? If not, the authors should consider those variables as covariates in the regression models. Also, in table 3, the value of “SV´SO” was missing.

RESPONSE: Thank you for this comment. As per your suggestion, we entered age and BMI as covariates in the regression models and updated the relevant text. Please refer to Page 8, where we highlighted the changes using yellow color. Given that we tested the moderating role of gender in the mediation model in the present study, gender was not entered as a covariate in the analyses. As for the moderated mediation model, according to your comment, we carefully checked the results in Table 3. In this manuscript, we examined the moderating role of self-objectification in the relationships between selfie-viewing and appearance comparisons and between appearance comparisons and facial dissatisfaction. Thus, in Table 3, we presented the interaction between selfie-viewing and self-objectification on appearance comparisons and the interaction between appearance comparisons and self-objectification on facial dissatisfaction. The value of “selfie-viewing × self-objectification” on facial dissatisfaction was not presented.

Final, there are a number of wording mistakes that need to be corrected, especially in the method and result section. The authors should carefully proofread the paper. Following are some examples:

P6, selfie-viewing:

“On item” --> one item;

“measuring these times” --> measuring these items;

“different response ranges are adopted” --> different response ranges were adopted

RESPONSE: Thank you for this comment. We really appreciate your time in reading our manuscript carefully. As per your suggestion, we read through the whole body of the manuscript and made changes in false spelling and tense usage. Changes were highlighted using yellow color.

Round 2

Reviewer 2 Report

Regarding the conceptualization of self-objectification, I suggest that the authors change their major mentions of “self-objectification” to “trait self-objectification”. Also, the authors should explicitly indicate that the self-objectification measurement used in the study was a trait measurement. One side issue is that, after the revision, the trait self-objectification was not clearly defined in the manuscript. How does the objectification theory define self-objectification as a stable personal trait? Is there any evidence supporting self-objectification as a trait?

Additionally, comparing the revised and old version, I don’t see much improvement on explaining the tripartite influence model. How does the tripartite influence model define media and peers? How are media and peers defined in the tripartite model related to selfie-viewing on social media? How does the model link media and peers to body dissatisfaction by proposing appearance comparison as the mediator? I don’t think we can assume all readers understand and are familiar with answers to those questions.

There are also some typos in the manuscript. The authors should do carefully a proofread.